# Decreased Fatty Acid Transporter FABP1 and Increased Isoprostanes and Neuroprostanes in the Human Term Placenta: Implications for Inflammation and Birth Weight in Maternal Pre-Gestational Obesity

**DOI:** 10.3390/nu13082768

**Published:** 2021-08-12

**Authors:** Livia Belcastro, Carolina S. Ferreira, Marcelle A. Saraiva, Daniela B. Mucci, Antonio Murgia, Carla Lai, Claire Vigor, Camille Oger, Jean-Marie Galano, Gabriela D. A. Pinto, Julian L. Griffin, Alexandre G. Torres, Thierry Durand, Graham J. Burton, Fátima L. C. Sardinha, Tatiana El-Bacha

**Affiliations:** 1Laboratory of Nutritional Biochemistry, Institute of Nutrition Josué de Castro, Federal University of Rio de Janeiro, Rio de Janeiro 21941-902, Brazil; livia.belcastro@gmail.com (L.B.); marcelle.a.saraiva@gmail.com (M.A.S.); danimucci@gmail.com (D.B.M.); 2LeBioME-Bioactives, Mitochondria and Placental Metabolism Core, Institute of Nutrition Josué de Castro, Federal University of Rio de Janeiro, Rio de Janeiro 21941-902, Brazil; ferreira_c.s@hotmail.com (C.S.F.); gabidap@gmail.com (G.D.A.P.); torres@iq.ufrj.br (A.G.T.); 3Department of Biochemistry, University of Cambridge, Cambridge CB2 1QW, UK; a-murgia@hotmail.it (A.M.); julian.griffin@imperial.ac.uk (J.L.G.); 4Department of Environmental and Life Sciences, University of Cagliari, 09124 Cagliari, Italy; carla.lai82@gmail.com; 5Institut des Biomolécules Max Mousseron (IBMM), UMR 5247, Université de Montpellier, CNRS, ENSCM, Bâtiment Balard, 1919 Route de Mende, 34293 Montpellier, France; claire.vigor@umontpellier.fr (C.V.); camille.oger@umontpellier.fr (C.O.); jean-marie.galano@umontpellier.fr (J.-M.G.); thierry.durand@umontpellier.fr (T.D.); 6Department of Metabolism, Digestion and Reproduction, Imperial College London, London SW7 2BX, UK; 7Lipid Biochemistry and Lipidomics Laboratory, Institute of Chemistry, Federal University of Rio de Janeiro, Rio de Janeiro 21941-598, Brazil; 8Centre for Trophoblast Research, Department of Physiology, Development and Neuroscience, University of Cambridge, Cambridge CB2 3EG, UK; gjb2@cam.ac.uk

**Keywords:** maternal pre-gestational obesity, placenta, lipid metabolism, fatty acid transporter proteins, isoprostanoids, neuroprostanes, isoprostanes, docosahexaenoic acid, arachidonic acid

## Abstract

The rise in prevalence of obesity in women of reproductive age in developed and developing countries might propagate intergenerational cycles of detrimental effects on metabolic health. Placental lipid metabolism is disrupted by maternal obesity, which possibly affects the life-long health of the offspring. Here, we investigated placental lipid metabolism in women with pre-gestational obesity as a sole pregnancy complication and compared it to placental responses of lean women. Open profile and targeted lipidomics were used to assess placental lipids and oxidised products of docosahexaenoic (DHA) and arachidonic acid (AA), respectively, neuroprostanes and isoprostanes. Despite no overall signs of lipid accumulation, DHA and AA levels in placentas from obese women were, respectively, 2.2 and 2.5 times higher than those from lean women. Additionally, a 2-fold increase in DHA-derived neuroprostanes and a 1.7-fold increase in AA-derived isoprostanes were seen in the obese group. These changes correlated with a 70% decrease in placental FABP1 protein. Multivariate analyses suggested that neuroprostanes and isoprostanes are associated with maternal and placental inflammation and with birth weight. These results might shed light on the molecular mechanisms associated with altered placental fatty acid metabolism in maternal pre-gestational obesity, placing these oxidised fatty acids as novel mediators of placental function.

## 1. Introduction

Maternal pre-gestational obesity and excessive gestational weight gain affect short- and long-term health of both the mother and her child [1]. Gestational diabetes mellitus and pre-eclampsia are complications of pregnancy associated with gestational obesity, and newborns from obese women have an increased risk of overgrowth. Obesity in adults and in children has reached epidemic proportions in Brazil [2] and worldwide [3]. Therefore, the rise in prevalence of obesity in women of reproductive age in both developed and developing countries might propagate intergenerational cycles of detrimental effects on metabolic health, contributing to substantial economic burden on society [4,5,6,7] and highlighting the necessity of determining the mechanisms involved.

Fatty acids are essential for the accretion of body fat in the fetus, especially during the last trimester of pregnancy. Long chain polyunsaturated fatty acids, in particular docosahexaenoic acid (DHA) and arachidonic acid (AA), have specific roles in membrane composition and the development of the retina, and are the major components of the white matter of the brain. Therefore, they are of paramount importance to proper neural and visual development and cognitive function [8,9,10,11]. Ex vivo placental perfusion data [12] and in vivo kinetics of ^13^C-fatty acids [13] show that maternal–fetal ^13^C-labelled lipid transfer is very low (1–6%), implying that lipid metabolism and handling by the placenta are strictly controlled and that these mechanisms are possibly major players in the allocation of fatty acids to fetal organs [14,15].

The inflammatory milieu imposed by maternal obesity disrupts the cross-talk between maternal signals and the placenta, resulting in impaired placental function [16]. Uptake and metabolism of essential fatty acids, particularly DHA, is impaired in placentas from obese women [17,18]. Increased accumulation of lipids in the placenta has been described in gestational obesity, which was associated with decreased oxidation of fatty acids and impaired mitochondrial function [19,20]. Conversely, other studies have reported that placental total lipid content [21] and maternal–fetal transfer of ^13^C-labelled non-essential fatty acids [13] is similar between lean and obese mothers. Altogether, these observations emphasize that placental lipid handling and metabolism might be disrupted by maternal obesity and deserve further investigation to clarify the mechanisms involved.

The maternal circulation, and ultimately the maternal diet, are the sources of polyunsaturated fatty acids, as their synthesis by the fetus and the placenta is limited and insufficient to meet the high demand imposed by the growing fetus. Placental fatty acid handling relies on several proteins which are responsible for (a) the uptake of fatty acids from the maternal circulation, partly as lipoproteins and as non-esterified fatty acids and lysophospholipids bound to albumin, and (b) the numerous metabolic fates of fatty acids and also their transfer to the fetus [22]. The hydrolysis of fatty acids from triacylglycerol in lipoproteins is catalyzed by endothelial lipase and by lipoprotein lipase. The former is selective for hydrolyzing unsaturated fatty acids esterified in the *sn*-2 position of glycerol. Fatty acids are then taken up by the placenta by fatty acid translocators (FAT/CD36), fatty acid transport proteins (FATP/SLC27A), and Mfsd2a. In the cytoplasm of the syncytiotrophoblast, fatty acid binding proteins (FABPs), which are noncatalytic binding proteins, mediate fatty acid metabolism and inflammatory processes [23].

Both the expression and the content of placental fatty acid transport proteins are altered by maternal obesity [19,24,25,26]. However, how these changes affect placental fatty acid metabolism and signalling properties, and fatty acid transport and availability to the fetus in maternal gestational obesity, is not fully understood. Oxidative stress and inflammation have been associated with enhanced contents of oxidised fatty acids and trophoblast dysfunction in pre-eclampsia. In particular, hydroxyeicosatetraenoic acids (HETEs), products of AA oxidation catalyzed by lipoxygenases and CYP, and F_2_-isoprostanes, products of non-enzymatic peroxidation of AA, are increased in placentas from pregnancies complicated by pre-eclampsia [27,28]. All these fatty acid metabolites have some degree of vasoconstrictive and pro-inflammatory effects and so may affect placental function.

Here, we characterized the major categories and classes of lipids in term placentas from pregnancies complicated by maternal pre-gestational obesity. We also analysed fatty acid transport proteins involved in the handling of polyunsaturated fatty acids by the placenta. A novel aspect of this study was a thorough characterization of placental non-enzymatically oxidised isoprostanoids derived from AA and DHA, which as inflammatory mediators might be a mechanistic link between pre-gestational obesity and placental dysfunction. Additionally, we assessed how polyunsaturated fatty acids and their oxidised fatty acids metabolites are associated with inflammation and lipid handling by the placenta, and possibly with neonatal outcomes in maternal pre-gestational obesity.

## 2. Materials and Methods

### 2.1. Study Design and Participants

The current study was part of a randomized controlled trial registered on clinicaltrials.org (NCT03215784), which was designed to evaluate the effects of fish oil and probiotics supplementation throughout pregnancy on women with pre-gestational obesity. The study was conducted at the *Maternidade Escola*, between January 2015 and July 2017. This is a referral hospital, belonging to the Federal University of Rio de Janeiro, dedicated to providing pre-natal and delivery care and puerperal consultations to the local community. Women were recruited up to 13 weeks of pregnancy and the inclusion criteria were age between 19–35 y, pre-gestational body mass index (BMI) between 18.5 and 24.9 kg/m^2^ (lean) or ≥30 and ≤40 kg/m^2^ (obesity class 1 and class 2; [29]), absence of pre-existing infectious or chronic disease, except for obesity, a single fetus, and non-smoker. As a part of the pre-natal care, women presenting with high-risk pregnancies received nutritional counselling. Dietary intake was obtained during consultations in the 3rd trimester by trained professionals (Appendix A).

Gestational weight gain was classified accordingly [30] and gestational outcomes and newborn information (weight and length at birth) were obtained from medical charts and classified according to the INTERGROWTH-21st Project Curves [31].

To reduce possible confounders, we opted to only include samples from women who did not develop gestational diabetes mellitus, pre-eclampsia, or other pregnancy complications. Hence, maternal pre-gestational obesity was the sole associated complication. Matched maternal blood, placental samples, and umbilical cord blood of 12 women (6 lean and 6 obese) were used.

### 2.2. Ethics

Women read and signed a ‘Free and Informed Consent’ form upon recruitment, and this study was approved by the local Ethics committee of the *Maternidade Escola* and by the National Ethics committee (approval number CEP: 34611513.0.0000.5257 on 14 October 2014).

### 2.3. Biological Samples

Maternal blood was collected at the 36th gestational week and placental and umbilical cord blood samples were collected at delivery. The time between delivery and collection was up to 20–30 min. Umbilical cord blood was obtained by venipuncture and placental tissue was collected according to established procedures [32]. Placental samples were taken from the maternal surface, placed into cryovials, frozen immediately in liquid nitrogen, and stored at −80 °C until analysis. A detailed description of placenta sampling is provided in Appendix A.

### 2.4. Chemical and Reagents

All standards used in the targeted isoprostanoids analyses were synthetized in house, as published [33,34,35,36,37]. Internal standards: C19-16-F_1t_-PhytoP and C21-15-F_2t_-IsoP; External standards: isoprostanes 15-F_2t_-IsoP, 15-*epi*-15-F_2t_-IsoP, 5-F_2t_-IsoP, and 5-*epi*-5-F_2t_-IsoP, and neuroprostanes 10-F_4t_-NeuroP, 10-*epi*-10-F_4t_-NeuroP, and 4(*RS*)-4-F_4t_-NeuroP.

### 2.5. Fatty Acid Transporter Proteins in Placental Tissue

#### 2.5.1. Quantitative Real-Time PCR (qPCR)

qPCR was performed using TaqMan™ Universal PCR Master Mix (Applied Biosystems, Thermo Fisher Scientific, Waltham, MA, USA) with the following primers from TaqMan™ Applied Biosystems, Thermo Fisher Scientific, Waltham, MA, USA: Hs00195812 (EL), Hs00155026 (FABP1), and Hs00997360 (FABP3). Glyceraldehyde-3-phosphate dehydrogenase (GAPDH) was used as the housekeeping gene and transcripts were calculated using the threshold cycle 2-ΔΔCT method [38]. A detailed description of qPCR analysis is available in Appendix A.

#### 2.5.2. Western Blotting

Primary antibodies for EL (MBS2013720) from My BioSource, Inc. (San Diego, CA, USA), FABP1 (AB7366), and FABP3 (AB16916) from ABCAM (Cambridge, United Kingdom) were used in the following dilutions, respectively: 1:1000 EL, 1:500, and 1:500. Loading control was performed with anti-β-actin antibody (Sigma-Aldrich, Saint Louis, MO, USA, SAB5500001). A full description of Western blotting analysis is provided in Appendix A.

### 2.6. Cytokines in Plasma and Placental Tissue

Quantification of the cytokines (IL-1β: Interleukin-1β; IL-6: Interleukin-6; IL-10: Interleukin-10; TNF-α: Tumour Necrosis Factor-α) in maternal plasma, umbilical cord plasma, and placental tissue protein extract was performed by the Luminex xMAP (Multiple Analytic Profiling, Thermo Fisher Scientific, Waltham, MA, USA) assay [39] using 100 μg of placental extract as described for Western blotting analysis.

### 2.7. Maternal Lipoprotein Profile

Triglycerides, cholesterol, low-density lipoprotein (LDL) and high-density lipoprotein (HDL) were determined by enzymatic assay commercial kits (Sigma-Aldrich, Saint Louis, MO, USA), and very-low density lipoprotein (VLDL) was estimated using the formula proposed by Friedewald, Levy and Fredrickson (1972) [40].

### 2.8. Ion Mobility QTOF LC/MS Lipid Profile Analysis of Placenta Samples

Lipids were extracted using a modified Folch method [41]. The non-polar extract in chloroform was dried under nitrogen and suspended in isopropanol:acetonitrile:water (IPA:ACN:H_2_O, 2:1:1, *v*/*v*/*v*) containing 25 deuterated lipids used as internal standards (representatives of phosphatidic acid, phosphatidylcholines, phosphatidylethanolamines, glycerophospholipids, phosphatidylinositols, phosphatidylserine, sphingomyelin, ceramides, triacylglycerols, and free fatty acids; Appendix A). All samples were analysed in positive and negative modes.

An Agilent 6560 Ion Mobility Quadrupole Time-of-Flight (IM-QTOF) mass spectrometer coupled with an Agilent 1290 UHPLC system was used to combine separation power and selectivity of LC, IM, and MS techniques. The Dual Agilent Jet Stream Electrospray Ionization Source was operated separately in positive and negative ion modes. The detailed description of the LC-MS analyses can be found in Appendix A.

#### Chromatogram Pre-Processing and Lipid Annotation

Data pre-processing, including mass and CCS re-calibration and feature finding, was carried out using the packages IM-MS Reprocessor, IM-MS Browser and Mass Profiler from the MassHunter suite vB.08.00 (Agilent Technologies, Santa Clara, CA, USA).

The resulting data matrices were processed using a KNIME pipeline comprising both KNIME native nodes and integrated R scripts. Feature annotation was performed based on the AccurateMassSearch node of the OpenMS library [42].

### 2.9. Isoprostanoids in Placental Tissue

Isoprostanoids in placenta were determined based on a microLC–MS/MS method [43] after lipid extraction by the Folch method [41], which were then mixed with a mixture of internal standards, followed by alkaline hydrolysis. MS analysis was performed in an AB Sciex QTRAP 5500 (AB Sciex, Framingham, MA, USA) with an electrospray ionization source operated in negative mode. Quantification of isoprostanoids was performed with the MultiQuant 3.0 software using specific internal standards. The detailed description of isoprostanoids analysis by microLC–MS/MS is described in Appendix A.

### 2.10. Statistical Analyses

Variables’ frequency distribution was assessed by standardized coefficients of skewness and kurtosis. Those with values < −2.0 or > +2.0 were characterized as having a non-normal distribution, and were then presented as median and interquartile interval, and compared using the Mann–Whitney test. Groups’ frequency distributions were compared using the chi-square test. Associations between continuous variables were assessed by Spearman’s correlation analysis. Stepwise multiple regression analyses (backward) were used to investigate the effect of independent factors on birth weight and on mothers and placental tissue content of isoprostanoids. The criteria for the inclusion of independent variables in the multiple regression models were based on results from Spearman correlations and on biochemical soundness. In the final model, only significant variables that improved the adjustment of the model were kept (*p*-to-remove ≥ 0.05; *p*-to-persist < 0.05). The multiple regression models were further assessed by analysis of residual plots that were checked to determine if they were randomly distributed. Data analyses were performed with GraphPad Prism v7.0 (GraphPad Software, San Diego, CA, USA) and Statgraphics Centurion v18 (Statgraphics Technologies, Inc.; The Plains, VA, USA). In all analyses, *p* < 0.05 was considered for rejection of the null hypothesis.

## 3. Results

### 3.1. General Characteristics of the Mothers and Newborns

The general characteristics of the mothers included in this study, their gestational outcomes and newborn information are presented in Table 1. Maternal pre-gestational BMI was significantly higher in the obese group (*p* < 0.05) and the median BMI value indicates obesity class I. Additionally, gestational weight gain was on average 40% lower in the obese mothers than in the lean group. No differences were observed in all other measurements, including gestational week at delivery, delivery mode and placental efficiency (weight at birth:placental weight ratio). Newborn outcomes were also similar despite the higher pre-gestational BMI in the obese group. Dietary records showed that there were no significant differences in energy and carbohydrate, protein, and lipid intake between groups during pregnancy (Appendix A). Therefore, the lower gestational weight gain observed in obese women may be explained, in part, as a consequence of nutritional counselling.

### 3.2. Placental Lipid Profile Suggests Alterations in Long-Chain Polyunsaturated Fatty Acids Abundance despite No Apparent Signs of Inflammation and Dyslipidemia in Maternal Pre-Gestational Obesity

Annotated species were separated into the lipid categories glycerophospholipids, sphingolipids, fatty acyls, glycerolipids, and sterols [44]. In both groups, glycerophospholipids was the most abundant category corresponding to 70% of the annotated signals, followed by sphingolipids (17%), fatty acyls (8%), glycerolipids (5%), and sterols (< 1%) (Figure 1A). No differences were observed in any lipid categories between the lean and obese women. Glycerophospholipids, sphingolipids, and glycerolipids were divided into classes. The most abundant classes with biological significance are shown in Figure 1B. Total glycerophospholipid was similar between the groups. However, the content of lysophospholipids tended to a marginal (*p* = 0.056) 1.25-fold increase in placentas from obese women. Ceramides and sphingomyelins were the most abundant sphingolipids that showed similar contents between groups. Lastly, contents of monoacylglycerol, diacylglycerol, and triacylglycerol did not differ between the groups.

Despite similar contents of fatty acyls (Figure 1A), some important long-chain *n*-3 polyunsaturated fatty acids varied between the groups (Figure 1C). Eicosapentaenoic acid (EPA) and DHA abundance were, respectively, 1.74 and 2.20 times higher (*p* < 0.05) in the obese group. Additionally, the mean value of AA was 2.5-fold higher in the obese group, although the *p*-value did not reach statistical significance. Changes in free fatty acids were restricted to essential polyunsaturated fatty acids, as contents of palmitic and stearic (saturated) and of palmitoleic and oleic (monounsaturated) acids were similar in the lean and obese women (Figure 1D).

Maternal obesity is often associated with an inflammatory milieu and dyslipidemia. Hence, the levels of IL-1 β, IL-6, IL-10, and TNF-α were measured in maternal blood, placental tissue, and umbilical cord blood (Appendix A). No differences were found in any of these inflammatory markers, except for a lower content of IL-1 β in placentas from women with pre-gestational obesity. Likewise, the content of total cholesterol, triacylglycerols, and HDL, LDL, and VLDL lipoproteins in maternal plasma were similar between lean and obese women (Appendix A).

These results suggest that placentas from women with pre-gestational obesity did not present signs of overall lipid accumulation and no apparent signs of inflammation nor maternal dyslipaemia were observed. However, placental polyunsaturated long-chain fatty acids were increased.

### 3.3. Fatty Acid Transporter Protein FABP1 Is Decreased and Negatively Associated with Polyunsaturated Fatty Acids in Placentas from Women with Pre-Gestational Obesity

We next analysed the fatty acid transporter proteins that handle unsaturated fatty acids by the placenta: endothelial lipase (EL), fatty acid binding protein-1 (FABP1), and fatty acid binding protein-3 (FABP3), at the protein and mRNA levels (Figure 2A,B). Placentas from the obese group presented a significant 70% decrease in FABP1 protein content compared with placentas from the lean group (*p* < 0.05) and a non-significant 30% decrease in mRNA levels. Endothelial lipase and FABP3 protein and transcript levels were similar between groups (Figure 2A,C).

The associations between the changes in FABP1 and polyunsaturated fatty acids were assessed by Spearman rank correlations, and the ratios of fatty acid (FA) to FABP1 were also assessed (Figure 2C–G). For all *n*-3 fatty acids (Figure 2C–E) and *n*-6 fatty acids (Figure 2F,G), the ratio FA:FABP1 was significantly higher in placentas from the obese group. The ratios EPA:FABP1 (Figure 2E), DHA:FABP1 (Figure 2F), and AA:FABP1 (Figure 2G) were, respectively, 5.5-, 7-, and 6-fold higher in placentas from the obese compared to the lean group. Spearman correlation analyses also showed that coefficients were negative for all FA with statistically significant results for EPA (*r* = −0.82; *p* = 0.02; Figure 2E) and marginally significant for DHA (*r* = −0.57; *p* = 0.09; Figure 2F) and AA (*r* = −0.60; *p* = 0.054; Figure 2G). Taken together, these results suggest that FABP1 might play a role in placental handling of these fatty acids and that the decrease in its content might have an impact in the availability and signalling properties of EPA, DHA, and AA, in particular.

### 3.4. Neuroprostanes and Isoprostanes Are Increased and Negatively Associated with FABP1 Protein in Placentas from Women with Pre-Gestational Obesity

MicroLC–MS/MS targeted analysis identified seven products of non-enzymatic peroxidation of the essential polyunsaturated fatty acids DHA and AA in placental tissue. Three DHA-derived neuroprostanes, namely 10(*R*)-10-F_4t_-NeuroP, 10(*S*)-10-F_4t_-NeuroP, and 4(*RS*)-4-F_4t_-NeuroP, and four AA-derived isoprostanes, namely 15-*epi*-15-F_2t_-IsoP, 15-F_2t_-IsoP, 5(*RS*)-5-F_2t_-IsoP, and 5-F_2c_-IsoP (Figure 3A) were found. All neuroprostanes and isoprostanes were increased (*p* < 0.05) in placentas from women with pre-gestational obesity, except for 4(*RS*)-4-F_4t_-NeuroP, which was marginally increased (*p* = 0.07). The overall increase of each DHA-derived neuroprostane was close to 2-fold, and of each AA-derived isoprostane was near 1.7-fold. To investigate the combined behavior of these fatty acids derived from DHA and AA, the sums of neuroprostanes and isoprostanes were compared between groups, and the same significant increase in placentas from women with pre-gestational obesity was observed (Figure 3B).

We calculated the ratio of each neuroprostane and isoprostane to FABP1 (Figure 3C,E) and performed correlation analyses (Figure 3D,F, respectively) following the same rationale as used to investigate associations between DHA and AA with FABP1. Significantly higher ratios of neuroprostanes and isoprostanes to FABP1 were found in placentas from obese compared to lean women (*p* < 0.05). The neuroprostane to FABP1 ratios were nearly 6-fold higher, and the isoprostanes to FABP1 ratios were nearly 5-fold higher in placentas from obese women. Negative correlations were found between the sum of neuroprostanes and isoprostanes and FABP1, but only the latter reached statistical significance (*r* = −0.42; *p* = 0.17 for neuroprostanes, Figure 3D, and *r* = −0.66; *p* = 0.02 for isoprostanes, Figure 3F). Taken together, these results suggest that the decrease in FABP1 protein observed in placentas from women with pre-gestational obesity is, to some degree, associated with the increase in neuroprostanes and isoprostanes observed in these placentas.

The metabolic sources of DHA-derived neuroprostanes, in particular, and AA-derived isoprostanes are not fully known; likely candidates are free DHA and AA, or phospholipids, mainly phosphatidylcholine, enriched in these fatty acids [45]. To address this issue, we calculated the ratios of neuroprostanes to free DHA (Figure 3G) and DHA-enriched phosphatidylcholine species (Figure 3G,H), and of isoprostanes to free AA and AA-enriched phosphatidylcholine species (Figure 3I,J). No significant differences were observed in the ratios of neuroprostanes and isoprostanes to free FA between the lean and obese groups. On the other hand, marginally lower ratios were observed for DHA-enriched phosphatidylcholine species and AA-enriched phosphatidylcholine in placentas from the obese compared to the lean group. These results suggest that DHA and AA in phosphatidylcholine might be less susceptible to non-enzymatic peroxidation in placentas from women with pre-gestational obesity. Non-esterified DHA and AA seem to similarly contribute to the synthesis of neuroprostanes and isoprostanes, respectively, in placentas from lean and obese women.

### 3.5. DHA-Derived Neuroprostanes and AA-Derived Isoprostanes Are Positively Associated with Maternal Pre-Gestational BMI and Endothelial Lipase Protein; and DHA-Derived Neuroprostanes Only Are Negatively Associated with Inflammation and Birth Weight

Spearman rank correlation was used to further investigate the associations of DHA-derived neuroprostanes and AA-derived isoprostanes with inflammation and placental lipid handling. Pre-gestational BMI (Figure 4A,C), endothelial lipase protein (Figure 4C,D), maternal plasma TNF-α (Figure 4E,G), and placental TNF-α (Figure 4F,H) were considered as independent variables. AA-derived isoprostanes, but not DHA-derived neuroprostanes, significantly correlated with pre-gestational BMI (*r* = 0.60 and *p* = 0.04; *r* = 0.48 and *p* = 0.12, respectively). Additionally, neuroprostanes (*r* = 0.64; *p* = 0.03) and isoprostanes (*r* = 0.78; *p* = 0.01) significantly correlated with endothelial lipase protein. Correlations between inflammation markers and oxidised fatty acids showed significant negative associations between DHA-derived neuroprostanes and maternal (*r* = −0.81; *p* = 0.02) and placental (*r* = −0.61; *p* = 0.04) TNF-α levels (Figure 4E,F). In contrast, no significant correlations were found between AA-derived isoprostanes and inflammation markers, except for a marginally significant negative correlation with placental TNF-α (*r* = −0.51; *p* = 0.09, Figure 4H).

Multiple regression analysis was used to investigate the tentative predictors of placental neuroprostanes and isoprostanes and their association with birth weight (Table 2). The sum of placental DHA-derived neuroprostanes was predicted by pre-gestational BMI (β = 1.64 × 10^2^; positive association) and by placental TNF-α content, with negative association (β = −5.56 × 10^3^) (Table 2; model 1). The sole predictor of the sum of placental AA-derived isoprostanes was placental endothelial lipase protein (β = 1.21 × 10^4^; model 2). Most importantly, birth weight was significantly determined by pre-gestational BMI (β = 1.26 × 10^2^), gestational weight gain (β = 7.72 × 10^1^), and by the sum of placental neuroprostanes, with a negative association (β = −4.06 × 10^−1^) (Table 2, model 3).

Collectively, DHA-derived neuroprostanes might mediate a less inflammatory response and potentially prevent excessive neonatal weight gain, as opposed to AA-derived isoprostanes, which appeared related to increased maternal adiposity.

## 4. Discussion

In this study, we show that despite no overall features of lipid accumulation, placentas from women with pre-gestational obesity contain an increased content of polyunsaturated free fatty acids. This increase is associated with a decreased content of FABP1, a cytoplasmic protein that handles unsaturated fatty acids and is important for determining their biological fate. Additionally, higher contents of neuroprostanes and isoprostanes, which are products of non-enzymatic oxidation of DHA and AA, respectively, were observed in placentas from obese women. Correlation and multivariate analyses suggested that these oxidised fatty acids are associated with maternal and placental inflammation, and also with birth weight. To our knowledge, this is the first report to characterize DHA-derived neuroprostanes and AA-derived isoprostanes in the human placenta, and in particular to explore their role in maternal and neonatal outcomes in the context of obesity.

Obesity in general is associated with chronic low-grade inflammation and insulin resistance. Therefore, obese women are at higher risk of developing metabolic disturbances during pregnancy [1]. In this study, samples from women who developed gestational diabetes mellitus, pre-eclampsia, or other complications were excluded from the analysis. Therefore, one of the strengths of our study is the well characterized patient cohort and the homogeneity of placental samples which reduced possible confounders of obesity-associated comorbidities. The median BMI value of the obese women was 34.3, indicating they present obesity class 1, and only one out of six women presented with a BMI of 37.6 (obesity class II). Additionally, a non-significant 40% decrease in gestational weight gain was observed in obese women. This result is in agreement with previous reports showing that obese women gain less weight than lean women during pregnancy [30]. The mechanisms involved are not well established, but it might be a physiological response to compensate for the excess stored energy. In our study, we found similar energy and nutrient intake between lean and obese women, possibly as a result of the nutritional counseling they received during the pre-natal care. The fact that the obese women did not present with other metabolic complications might explain the similar plasma concentrations of inflammatory markers, and lipoproteins and triacylglycerols as in lean women. We speculate, therefore, that despite being obese, they were metabolically healthy, and this may explain the lack of differences in placental weight and efficiency. Although several studies have shown that maternal obesity is associated with an inflammatory and pro-oxidant milieu [46,47], there are reports of healthy obese mothers with no signs of dyslipidaemia [13,48]. Indeed, a lower content of IL-6 and reduced infiltration of macrophages in placentas from obese women was recently shown [49]. Conversely, there are consistent data showing that several placental responses appear to be affected by maternal pre-gestational obesity, irrespective of its severity [16,50].

The placenta is the highly specialized organ that interfaces between the mother and her baby. It integrates signals of maternal availability and fetal demand of nutrients and oxygen, and has a central role in determining the life-long health of the offspring [51]. The importance of placental lipid metabolism in regulating maternal-fetal transfer of fatty acids, in particular essential polyunsaturated fatty acids, has been recognized in healthy pregnancies [14,52] and in pre-gestational obesity [17,18], although the mechanisms involved are not fully known. Indeed, in this study, we showed that placentas from women with pre-gestational obesity contain a significant increase in DHA and EPA content compared to placentas from lean women. Additionally, the median value of placental AA from obese women was nearly 2.5-fold higher than the lean group, despite these differences being non-significant statistically, which may reflect the small sample size.

Alterations in placental fatty acid transporter proteins are plausible mechanisms associated with altered placental lipid handling in maternal pre-gestational obesity [53]. Indeed, we show that FABP1 is decreased in placentas from obese women, corroborating a previous study [25]. The decrease in FABP1 in placentas from the obese group was reflected in significantly higher ratios of *n*-3 and *n*-6 polyunsaturated fatty acids to FABP1.

FABP1 is the liver isoform of FABP. Unlike other members in the FABP family, it has two ligand-binding sites for fatty acids and higher affinity for long-chain unsaturated fatty acids. Besides fatty acids, FABP1 binds a range of hydrophobic molecules, such as peroxisome proliferator-activated receptors (PPARs), prostaglandins, hydroxyl and hydroperoxyl metabolites of AA, lysophopholipids, and pro-oxidants such as heme [54]. FABP1 has been found in the cytosol, nucleus, and mitochondria [55,56], and so may have multiple roles. Hence, we speculate that the decreased content of FABP1 found in placentas from women with pre-gestational obesity might have affected placental function beyond just lipid transport.

Due to the role of FABP1 in trafficking fatty acids to the nucleus and also its ability to bind PPARs, the decrease in placentas from women with pre-gestational obesity likely altered pathways related to placental lipid handling and fatty acid metabolism. There is evidence that PPARs are involved in these processes in the human placenta [57,58]. Similar contents of PPARs in placentas from obese and lean women [25] were described in parallel with increased PPAR-γ (related to fatty acid synthesis) and decreased PPAR-α (related to fatty acid oxidation) [20]. Yang et al. have proposed that regulation of PPARs is important at the maternal–fetal interface and this fact might explain the apparent discrepancies in PPARs content in maternal pre-gestational obesity [59]. It remains to be determined if and how PPARs participate in the cross-talk between long chain polyunsaturated fatty acids and FABP1 protein seen in the present study.

As FABP1 binds to pro-oxidant molecules and is considered an antioxidant, at least in the liver [54], its decrease might render the placenta more susceptible to oxidative stress in obese women. A decreased total antioxidant capacity and increased activation of the pro-oxidant NFkB pathway has been observed in placentas from women with obesity [60]. The significant increase in DHA-derived neuroprostanes and AA-derived isoprostanes we observed supports a pro-oxidant milieu in placentas from the obese group. In addition to increased concentrations of neuroprostanes and isoprostanes, we found that their ratios to FAPB1 were also significantly altered, and that FABP1 negatively correlated with the total isoprostanes. These results not only support the concept of a pro-oxidant milieu, but also suggest that FABP1 binds to isoprostanoids, in particular AA-derived isoprostanes, as observed for enzymatically derived fatty acid mediators.

Neuroprostanes and isoprostanes are isoprostanoids (isomers of prostaglandins), formed by the non-enzymatic peroxidation of DHA and AA, respectively. They have been described as important signalling molecules, and AA-derived isoprostanes act in vascular smooth muscle through tyrosine kinase and Rho kinase in human cells [61] and through prostanoid receptors in rats [62]. AA-derived isoprostanes are implicated in cardiovascular diseases acting as vasoconstrictors and considered markers of oxidative stress [45]. Conversely, recent evidence suggests that DHA-derived neuroprostanes have anti-inflammatory properties [63,64], despite the fact that they have been implicated in oxidative stress in neurodegenerative diseases [65]. Their role in placental function in pregnancies complicated by maternal obesity has not yet been described.

Oxidised 9-HODE, 13-HODE, and 15-HETE, products of AA produced by cycloxygenase activity, did not affect expression of syncytin, cyclin E, and p27, which are markers of trophoblast differentiation, indicating that they are not implicated in trophoblast dysfunction, at least in healthy trophoblasts [66]. On the other hand, in placentas from pregnancies complicated with pre-eclampsia, it has been shown that F_2_ class isoprostanes derived from AA contribute to oxidative stress and have vasoconstrictive properties [67].

The formation of isoprostanoids is regulated to some extent, and are predominantly formed in situ from oxidation of DHA or AA esterified to phosphatidylcholine in cell membranes. Therefore, their signalling properties depend upon phospholipase A_2_ activity [45], and a strict correlation between placental phospholipase A_2_ mRNA and free F_2_-isoprostane levels has been found in pre-eclampsia [67].

Our marginally lower ratios of neuroprostanes to DHA-enriched and of isoprostanes to AA-enriched PC in placentas from obese women suggest that the proportion of these mediators formed in situ is decreased in obesity. The significance of this result may be related to compensatory mechanisms in placentas from obese women, controlling membrane function and avoiding excessive release of mediators and possibly controlling neonatal birth weight. Indeed, increased placental phospholipase A_2_ activity has been correlated with increased placental accumulation of lipids and adiposity in the newborn [68].

In the present study, AA-derived isoprostanes correlated with maternal pre-gestational BMI. Additionally, in the multivariate model, the sole predictor of isoprostanoids in the placenta was placental EL protein. These results could be interpreted in the light of the pro-inflammatory effect of AA-derived isoprostanes. A higher content of EL has been described in placentas from pregnancies complicated with gestational diabetes mellitus, in addition to obesity [69]. Consistent with these findings, we observed a non-statistical 25% increase in EL in placentas from women with pre-gestational obesity.

In the case of neuroprostanes, multivariate models indicated that both maternal pre-gestational BMI (positive β coefficient) and placental TNF-alpha (negative β coefficient) were tentative predictors of their content in the placenta, suggesting their possible anti-inflammatory role. This association was also observed in the correlation analysis, where the sum of neuroprostanes was negatively correlated with maternal and placental TNF-α contents. Importantly, when we evaluated the possible involvement of neuroprostanes and isoprostanes with birth weight, total neuroprostanes presented a negative contribution and, as expected, pre-gestational BMI and gestational weight gain had positive associations. However, total isoprostanes did not fit the same experimental model. These results suggest that neuroprostanes present anti-inflammatory roles in placentas from women with pre-gestational obesity, and by negatively affecting birth weight might attenuate the intergenerational cycles of detrimental effects on metabolic health as the risk of overweight/obesity in childhood increases gradually over the full range of maternal pre-gestational BMI [70]. Indeed, neonatal birth weight was similar between groups in the present study.

A word of caution is necessary when investigating birth weight in the context of gestational obesity as it has been suggested that neonatal adiposity, and not total weight, is independently associated with childhood adiposity [71]. The association between placental oxidised fatty acids and neonatal outcomes, including adiposity, has never been addressed. An additional possible mechanism by which neuroprostanes exert anti-inflammatory effects is the observation that 4-(*RS*)-4-F_4t_-NeuroP increased mRNA levels of the enzyme heme-oxygenase, which degrades heme, in human neuroblastoma cells and in primary culture of neurons [65]. Given the fact that FABP1 is able to bind heme, a molecule with pro-oxidant properties, the increase in neuroprostane isomers might counteract the decrease in FABP1 protein in the obese group, providing a means of anti-oxidant defense. It remains to be determined if this is the case in the term human placenta in maternal pre-gestational obesity.

The main limitation of the present study is the sample size, which impacts on the statistical power. Consequently, results from the multiple regression analyses should be considered as an approach to screen for predictors in each model. In addition, we were not able to address sexually dimorphic responses in placental fatty acid metabolism. This aspect is important as a recent study showed that placentas from females appear more prone to store and esterify fatty acids, while placentas from males have a decreased capacity to transfer DHA to cord blood [72]. Nonetheless, our data are biochemically sound and bring new knowledge to the field, identifying isoprostanoids as potential novel mediators of placental function.

As maternal diet is the ultimate source of DHA and AA, our results add to the body of evidence that there must be a balance between intake of *n*-3 and *n*-6 fatty acids, and possibly a higher intake of DHA is necessary during pregnancy [73]. A recent pilot trial demonstrated that DHA supplementation starting early in pregnancy promoted higher lean mass accrual at birth and improved fetal growth [74]. The next step is to evaluate the intake of individual fatty acids and also to address the quality of the diet of these mothers by assessing other factors, e.g., dietary antioxidants, that contribute to placental lipid handling in a larger cohort. Intervention studies are also needed to identify the molecular mechanisms involved. Such data would form the basis of efficient dietary strategies to decrease the burden of maternal pre-gestational obesity.

## 5. Conclusions

The decreased content of FABP1 and increased content of DHA, EPA, and AA in placentas from women with pre-gestational obesity indicate important alterations in placental lipid metabolism and fatty acid handling even in obese women that did not present major signs of inflammation. Additionally, the decrease in FABP1 content was associated with increased contents of DHA-derived neuroprostanes and AA-derived isoprostanes in placentas from obese women. Distinct mechanisms seemed to contribute to their increased content, as total DHA-derived neuroprostanes was negatively associated with placental TNF-α, and AA-isoprostanes associated with pre-gestational BMI. Importantly, placental neuroprostanes negatively contributed to birth weight. Taken together, these results confirm previous observations showing that placental lipid metabolism is disrupted in maternal obesity. The possibility that these oxidised fatty acids act as mediators of placental, and possibly maternal, inflammation opens several possibilities for the investigation of the molecular mechanisms associated with alterations in placental fatty acid metabolism in maternal pre-gestational obesity and their possible role in the health of the offspring.

## Figures and Tables

**Figure 1 nutrients-13-02768-f001:**
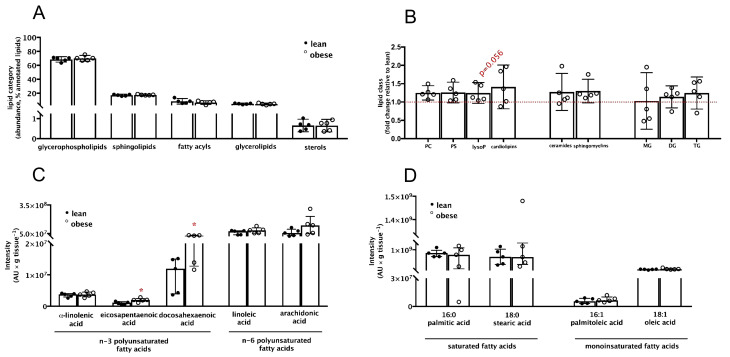
Placental lipid profile suggests alterations in long-chain polyunsaturated free fatty acids abundance in maternal pre-gestational obesity. Placental lipid profile was determined by IM-QTOF LC/MS after lipid extraction by the Folch method. Ion intensities were processed using a KNIME pipeline comprising both KNIME native nodes and integrated R scripts, and lipids were assigned into categories, classes, and species. Placentas from lean (●) and from obese (○) women were compared according to (**A**) the abundance of lipid categories expressed relative to total lipid annotated; (**B**) the major lipid classes in each category; and (**C**) the annotated polyunsaturated and (**D**) saturated and monounsaturated fatty acid species. PC: phosphatidylcholine; PS: phosphatidylethanolamine; lysoP: lysophospholipids; MG: monoacyglycerols; DG: diacyglycerols; TG: triacylglycerols. * Significantly different from the lean group; *p* < 0.05 (Mann–Whitney test).

**Figure 2 nutrients-13-02768-f002:**
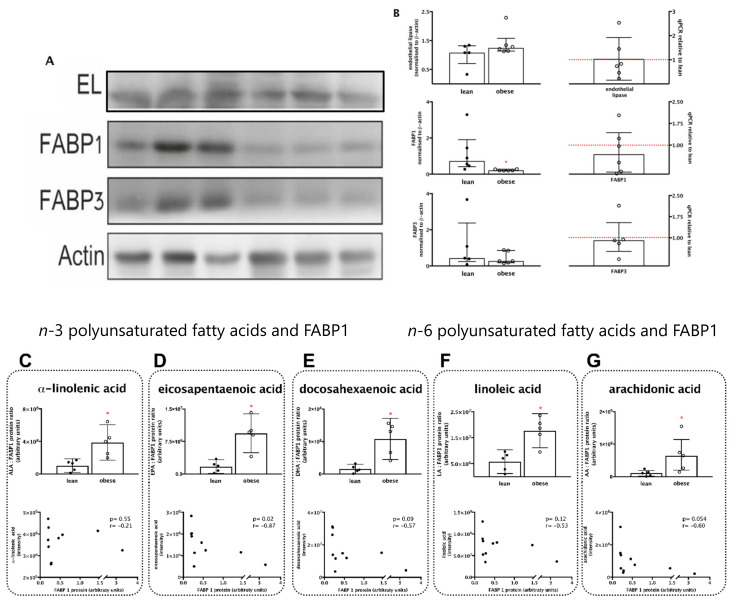
Fatty acid transporter protein FABP1 is decreased and negatively associates with long-chain polyunsaturated fatty acids in placentas from women with pre-gestational obesity. Placentas from lean (●) and from obese (○) women were compared according to placental fatty acid transporter proteins endothelial lipase, FABP1, and FABP3 at the protein and transcript levels (**A**,**B**). The ratio of polyunsaturated fatty acids to FABP1 and the respective Spearman correlations were calculated for the *n*-3 fatty acids α-linolenic (**C**), eicosapentaenoic (**D**), and docosahexaenoic (**E**), and for the *n*-6 fatty acids linoleic (**F**) and arachidonic (**G**), to investigate possible associations between alterations in FABP1 and long-chain fatty acid contents as a function of maternal pre-gestational obesity. * Significantly different from the lean group; *p* < 0.05 (Mann–Whitney test).

**Figure 3 nutrients-13-02768-f003:**
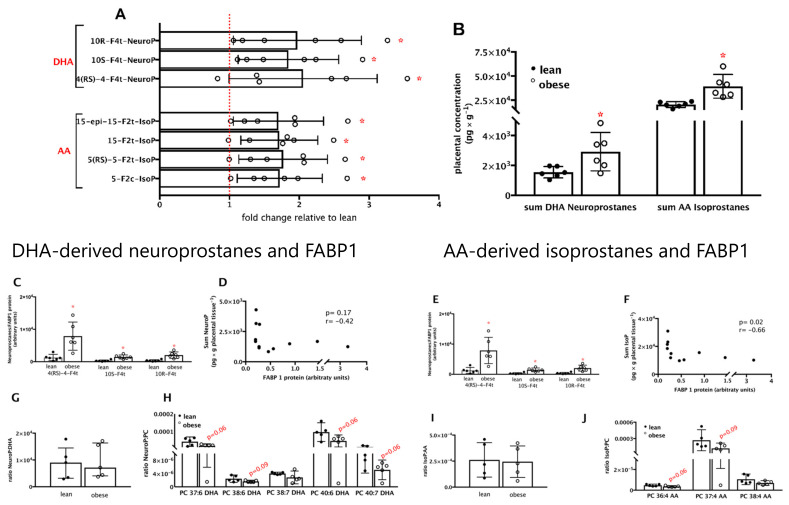
Neuroprostanes and isoprostanes are increased and negatively associated with FABP1 protein in placentas from women with pre-gestational obesity. Products of non-enzymatic peroxidation of the essential polyunsaturated fatty acids docosahexaenoic (DHA) and arachidonic (AA) in placental tissue were analyzed by target microLC–MS/MS. Three DHA-derived neuroprostanes (NeuroP) and four AA-derived isoprostanes (IsoP) were identified. Placentas from lean women (●) and from obese women (○) were compared to investigate differences in DHA- and AA-derived isomers in placentas from obese relative to lean women (**A**) and in the sum of neuroprostanes and isoprostanes (**B**). The ratio of neuroprostane isomers to FABP1 and the respective Spearman correlations (**C**,**D**), and the ratio of isoprostanes to FABP1 and the respective Spearman correlations (**E**,**F**), were calculated to investigate possible associations between alterations in FABP1 and DHA and AA non-enzymatic peroxidation as a function of maternal pre-gestational obesity. Possible differences in the sources of neuroprostanes and isoprostanes induced by maternal pre-gestational obesity were assessed by the ratios of neuroprostanes to free DHA (**G**) and to DHA-phosphatidylcholine species (PC; **H**), and isoprostanes to free AA (**I**) and to AA-phosphatidylcholine species (PC; **J**). * Significantly different from the lean group; *p* < 0.05 (Mann–Whitney test).

**Figure 4 nutrients-13-02768-f004:**
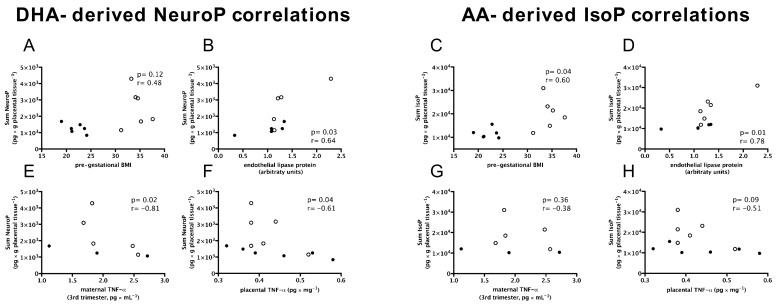
DHA-derived neuroprostanes are negatively correlated with inflammation markers, and AA-derived isoprostanes are positively associated with maternal pre-gestational BMI. Spearman rank correlations of DHA-derived neuroprostanes (NeuroP; **A**,**B**) and AA-derived isoprostanes (IsoP; **C**,**D**) and pre-gestational BMI and endothelial lipase, respectively, and DHA-derived neuroprostanes (**E**,**F**) and AA-derived isoprostanes (**G**,**H**) and maternal and plasma TNF-α, respectively. Placentas from lean women (●) and from obese women (○).

**Table 1 nutrients-13-02768-t001:** Clinical characterization of the mothers and newborns participating in the study.

Mothers		
	Lean (*n* = 6)	Obese (*n* = 6)
Maternal age (years) ^a^	26.0 (19.0–32.0)	26.5 (20.0–31.0)
Maternal pre-gestational BMI (kg/m^2^) ^a^	22.0 (19.0–24.2)	34.3 (31.2–37.6) *
Gestational weight gain (kg) ^a^	11.6 (10.1–21.3)	6.6 (1.5–19.0)
Gestational age (weeks)	40.5 (38.0–41.0)	39.0 (38.0–41.0)
Complications of pregnancy (other than pre-gestational obesity)	none	none
Delivery mode (*n*; %)	vaginal (2; 33); c/s ^B^ (4; 66)	vaginal (1; 17); c/s ^B^ (5; 83)
**Newborn**		
Placental weight (g) ^a^	485.0 (400.0–555.0)	480.0 (480.0–575.0)
Placental efficiency (birth weight:placental weight ratio)	6.9 (6.1–10.0)	6.9 (6.4–7.4)
Birth weight (kg) ^a^	3.3 (3.1–4.0)	3.4 (3.0–4.0)
Birth length (cm) ^a^	49.5 (46.0–53.5)	48.5 (45.0–52.0)

^a^ values expressed as median (minimum-maximum); * significantly different compared to lean women, *p* < 0.01, Mann–Whitney test; ^B^ c/s, cesarean section.

**Table 2 nutrients-13-02768-t002:** Multiple regression models for the assessment of predictors of placental Σ neuroprostanes and Σ isoprostanes and predictors of birth weight.

Dependent Variables	Independent Variables	β Coefficients			Adj. *R*^2^	Estimated Error (%) ^1^	*p* ^2^
		Value	SE	*p*-Value			
Σ Neuroprostanes in placenta **Model 1**	Pre-gestational BMI	1.64 × 10^2^	2.61 × 10^1^	0.0004	93.88	20.1	0.0000
	TNF-α, placenta	−5.56 × 10^3^	1.59 × 10^3^	0.0101			
	Gestational weight gain	–	–	ns	–	–	ns
	IL-6, placenta	–	–	ns	–	–	ns
	Endothelial lipase, protein	–	–	ns	–	–	ns
	FABP-1, protein	–	–	ns	–	–	ns
Σ Isoprostanes in placenta **Model 2**	Endothelial lipase, protein	1.21 × 10^4^	1.12 × 10^3^	0.0000	93.65	22.9	0.0000
	Pre-gestational BMI	–	–	ns	–	–	ns
	Gestational weight gain	–	–	ns	–	–	ns
	TNF-α, placenta	–	–	ns	–	–	ns
	IL-6, placenta	–	–	ns	–	–	ns
Birth weight **Model 3**	Pre-gestational BMI	1.26 × 10^2^	1.38 × 10^1^	0.0001	99.07	6.83	0.0000
	Gestational weight gain	7.72 × 10^1^	1.71 × 10^1^	0.0040			
	Σ Neuroprostanes, placenta	−4.06 × 10^−1^	1.37 × 10^−1^	0.0253			
	Σ Isoprostanes, placenta	–	–	ns	–	–	ns
	IL-6, placenta	–	–	ns	–	–	ns
	TNF-α, placenta	–	–	ns	–	–	ns
	Endothelial lipase, protein	–	–	ns	–	–	ns

^1^ Estimated relative error of estimate = (estimated absolute error × 100%)/average value of the dependent variable; ^2^ Model significance. Significant associations were independent of the following variables, denoted as non-significant (ns) *p*-values: model 1, gestational weight gain, IL-6 (placenta), endothelial lipase protein (placenta), and FABP1 (placenta); model 2, pre-gestational BMI, gestational weight gain, TNF-α (placenta), and IL-6 (placenta); model 3, Σ of isoprostanes (placenta), interleukin-6 (placenta), TNF-α (placenta), and endothelial lipase protein (placenta); – blank cell; IL-6: Interleukin-6; TNF-α: Tumour Necrosis Factor-α.

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
