# Peer review of "Decreased Fatty Acid Transporter FABP1 and Increased Isoprostanes and Neuroprostanes in the Human Term Placenta: Implications for Inflammation and Birth Weight in Maternal Pre-Gestational Obesity"

_nutrients, 2021, doi:10.3390/nu13082768_

Round 1
Reviewer 1 Report
This report provides new information on lipid biology of the placenta in the context of an obese pregnancy. That aspect of the presentation is good.
There were some specific points that should be clarified.
In their table, they indicate that there were no clinical conditions associated with obesity. It would be surprising if there were no cases of gestational hypertension or diabetes among the overweight women. Either condition could contribute or exacerbate the patterns they observed. There are many reports of placental pathophysiology in a diabetic pregnancy.
They don’t comment on any of the results in their Table that don’t differ between lean and obese women. For example, were they surprised that there was a significant difference in the placental weights and birthweights between the two groups. Wouldn’t they have expected to see larger babies born to the heavier women in keeping with what they wrote in the Introduction about the intergenerational effects?
They interpret their results as being a outcome of the proinflammatory state associated with pregnancy. However, it seems that they did not find any significant different in cytokine levels between the two groups – not in the maternal blood, placenta, or cord blood at delivery. They need to address this discrepancy in the Discussion, or else to not play up the role of inflammation in the Introduction.
They report that they measured 4 cytokines. It seems they may have opted to focus on IL-6, which is not necessarily bad. But many find that TNFalpha is a more sensitive cytokine to track in pregnant women if interested in individual differences.
Perhaps the link to the Supplemental figure with the cytokine results will become active alter. But right now the information provided to make it available does not work.
Lastly, they allude to the possible role of an obesogenic diet but don’t seem to present any information on food consumption. Do they know if the diets and/or caloric intake was similar between the two groups.
If they did not directly measure food consumption, a possible proxy might be to report preconception BMI and gestational weight gain. In many countries, overweight women are told by their OB/GYN to limit their weight gain, and to gain less weight than lean women. If they have information on GWG in the two groups, they could include it in their table with summary descriptives.
In sum, the report provides novel information on placental lipid metabolism.
Author Response
Please find below the responses to the reviewers’ comments to the manuscript “Decreased fatty acid transporter FABP1 and increased isoprostanes and neuroprostanes in the human term placenta: implications to inflammation and birth weight in maternal pre-gestational obesity”, by Livia Belcastro, Carolina S. Ferreira, Marcelle A. Saraiva, Daniela B. Mucci, Antonio Murgia, Carla Lai, Claire Vigor, Camille Oger, Jean-Marie Galano, Gabriela D.A. Pinto, Julian L. Griffin, Alexandre G. Torres, Thierry Durand, Graham J. Burton, Fatima L.C. Sardinha and myself.
The manuscript was thoroughly revised for English language by a native speaker. We would like to take the opportunity to thank the reviewers for their careful and thoughtful comments which certainly contributed to improve the quality of the manuscript.
Hoping to hear from you soon
Yours Sincerely,
Tatiana El-Bacha
Associate Professor
Institute of Nutrition, Federal University of Rio de Janeiro
Reviewer 1
This report provides new information on lipid biology of the placenta in the context of an obese pregnancy. That aspect of the presentation is good.
There were some specific points that should be clarified.
In their table, they indicate that there were no clinical conditions associated with obesity. It would be surprising if there were no cases of gestational hypertension or diabetes among the overweight women. Either condition could contribute or exacerbate the patterns they observed. There are many reports of placental pathophysiology in a diabetic pregnancy.
Response: We thank the reviewer for this comment and we would like to clarify that for this particular study we excluded samples from women who presented gestational diabetes and hypertensive disorders. Therefore, the only pregnancy complication presented by these women was pre-gestational obesity, which contributes to the homogeneity of placental samples to reduce possible confounders of obesity-associated comorbidities. In this sense, the increased contents of oxidised fatty acids (i.e., neuroprostanes and isoprostanes) is suggestive of placental responses due to gestational obesity.
This information has been better described in the methods section (lines 151-153), and commented on in the second paragraph (lines 545-551) of the discussion.
They don’t comment on any of the results in their Table that don’t differ between lean and obese women. For example, were they surprised that there was a significant difference in the placental weights and birthweights between the two groups. Wouldn’t they have expected to see larger babies born to the heavier women in keeping with what they wrote in the Introduction about the intergenerational effects?
Response: We thank the reviewer for this observation and we included comments of this nature in the discussion (lines 559-565; 698-702). There are a couple of aspects that might explain the lack of differences mentioned by the reviewer. The first one relates to the nature of this cohort, which was commented on above. Women who developed metabolic complications during pregnancy were excluded, which probably minimized placental overgrowth. Another aspect that should be mentioned is the small sample size of our cohort. In larger cohorts, increased birth weight is observed in pregnancies complicated by obesity and its associated comorbidities. However, when it comes to addressing intergenerational effects, neonatal adiposity seems to be independently associated with childhood adiposity, and not necessarily increased birthweight. This finding explains the fact that obesity is likely to develop postnatally. This information was addressed in the discussion of the revised manuscript and the following reference was included [72, of the revised manuscript]: Josefson JL, Scholtens DM, Kuang A, Catalano PM, Lowe LP, Dyer AR, Petito LC, Lowe WL Jr, Metzger BE; HAPO Follow-up Study Cooperative Research Group. Newborn Adiposity and Cord Blood C-Peptide as Mediators of the Maternal Metabolic Environment and Childhood Adiposity. Diabetes Care. 2021 May;44(5):1194-1202.
They interpret their results as being a outcome of the proinflammatory state associated with pregnancy. However, it seems that they did not find any significant different in cytokine levels between the two groups – not in the maternal blood, placenta, or cord blood at delivery. They need to address this discrepancy in the Discussion, or else to not play up the role of inflammation in the Introduction.
Response: We thank the reviewer for this comment and this aspect has been better addressed in the discussion (lines 559-572 and an additional reference number 49) and conclusion (lines 741-744) of the revised manuscript.
Regarding the interpretation of our results as a consequence of the proinflammatory state we would like to clarify this point: despite no differences in the levels of cytokines between lean and obese women, correlation and multiple regression analyses using as independent variables TNF-ain the placenta and TNF-ain maternal blood and BMI, showed important significant results with placental content of neuroprostanes and isoprostanes, as follows: we interpreted the increase in neuroprostanes as a placental response possibly associated with a less inflammatory state, taken the negative correlation with maternal (Figure 4E) and placental TNF-alevels (Figure 4F) and multiple regression analysis (Table 2; model 1). On the other hand, we interpreted the increase in isoprostanes as a placental response possibly associated with a more inflammatory state, taken their positive correlation with pre-gestational BMI (Figure 4C).
Additionally, we do not see the lack of significant difference in cytokine levels between groups as being a discrepancy. The fact that obese women did not present other metabolic complications supports these results, as commented previously, suggesting they were metabolically healthy.
They report that they measured 4 cytokines. It seems they may have opted to focus on IL-6, which is not necessarily bad. But many find that TNFalpha is a more sensitive cytokine to track in pregnant women if interested in individual differences.
Response: We would like to reinforce that our results indicated that there were no differences in the levels of cytokines (TNF-a, IL-6, IL-1, IL-10) between lean and obese women. And these results were shown as supplementary Figure 1. Unfortunately, the reviewer seemed not to have been able to access this file (maybe the editorial office might assist the reviewer with that matter). We used Spearman rank correlation to investigate possible associations between these inflammatory markers and placental oxidised fatty acids (Figure 4). This approach indicated that despite no absolute differences, TNF-alevels both in maternal blood and in placental tissue were negatively associated with neuroprostane content. Any other cytokine was significantly correlated with placental fatty acids. Therefore, we suggested that the higher content of neuroprostanes in placentas from obese women might indicate a less inflammatory response, considering the pro-inflammatory role of TNF-a. In the revised manuscript, we address this aspect to clarify any misleading information (conclusion lines 741-744).
Perhaps the link to the Supplemental figure with the cytokine results will become active alter. But right now the information provided to make it available does not work.
Response: That is unfortunate you could not open the supplemental material.
Lastly, they allude to the possible role of an obesogenic diet but don’t seem to present any information on food consumption. Do they know if the diets and/or caloric intake was similar between the two groups.
Response: We thank the reviewer for addressing this aspect. Dietary records were collected and nutritional counselling was provided to pregnant women as a part of their prenatal care. A detailed analysis of their dietary intake will be the focus of another manuscript, and that is the reason we opted to not include dietary intake data in the present study. That said, we agree with the reviewer regarding the importance of dietary records to help us interpreting the results and we included a table with energy and carbohydrate, protein and lipid intake (Supplementary table 2) in the revised manuscript. There were no significant differences in energy or macronutrient intake between the groups. Therefore, altered placental metabolism is unlikely to be due to major differences in their diet. Information of this nature has been included in the Supplementary Methods; Supplementary table 2; Results (lines 331-335) and discussion (lines 557-559 and 725-730) of the revised manuscript.
If they did not directly measure food consumption, a possible proxy might be to report preconception BMI and gestational weight gain. In many countries, overweight women are told by their OB/GYN to limit their weight gain, and to gain less weight than lean women. If they have information on GWG in the two groups, they could include it in their table with summary descriptives.
Response: Referring to gestational weight gain, this information is included in Table 1 of the original manuscript. We understand the point raised by the reviewer and for the sake of clarity we better addressed gestational weight gain in the discussion of the revised manuscript (discussion lines 553-559).
Gestational weight gain was, on average,40 % lower in obese women (non-significant). This result is in agreement with previous reports that showed that obese women gain less weight than lean women during pregnancy. As pointed previously, we found similar energy and nutrient intake between lean and obese women, possibly as a result of nutritional counseling they received during the pre-natal care which might explain, in part, the decreased gestational weight gain.
In sum, the report provides novel information on placental lipid metabolism.
Response: We thank the reviewer for the careful reading of the manuscript and for the insightful comments.
Reviewer 2 Report
This was a very small observational study in 6 obese and 6 lean and pregnant woman.
It was a difficult paper to wade through as it is so long and there is a lot of repetition, I found it a real struggle and found the lack of focus made it difficult to engage with this paper. Your readers are unlikely to read this paper unless you make this shorter and concentrate on your message. Methodology was repeated in the results and justification of tests was in the methods etc. I strongly recommend authors reduce the length of the paper; maybe move some of the methodology to a supplementary file, simplify the introduction, results and conclusion as the message is lost in the long paper. The discussion is very confusing as it flits from one topic to the next with no cohesion.
I think there are major limitations to this paper, the first is the sample size, the multiple variables measured (you only used multiple regression in a very small sample size - with so many variables; it can be a ‘shotgun regression’. I suggest you consider other analysis for example AIC ) , the fact that it is observational and no diet diaries are used. A RCT with supplementation/ no supplementation in lean and obese woman is needed to see if the results are due to intake or disordered metabolism.
Author Response
Please find below the responses to the reviewers’ comments to the manuscript “Decreased fatty acid transporter FABP1 and increased isoprostanes and neuroprostanes in the human term placenta: implications to inflammation and birth weight in maternal pre-gestational obesity”, by Livia Belcastro, Carolina S. Ferreira, Marcelle A. Saraiva, Daniela B. Mucci, Antonio Murgia, Carla Lai, Claire Vigor, Camille Oger, Jean-Marie Galano, Gabriela D.A. Pinto, Julian L. Griffin, Alexandre G. Torres, Thierry Durand, Graham J. Burton, Fatima L.C. Sardinha and myself.
The manuscript was thoroughly revised for English language by a native speaker. We would like to take the opportunity to thank the reviewers for their careful and thoughtful comments which certainly contributed to improve the quality of the manuscript.
Hoping to hear from you soon
Yours Sincerely,
Tatiana El-Bacha
Associate Professor
Institute of Nutrition
Federal University of Rio de Janeiro
Reviewer 2
This was a very small observational study in 6 obese and 6 lean and pregnant woman.It was a difficult paper to wade through as it is so long and there is a lot of repetition, I found it a real struggle and found the lack of focus made it difficult to engage with this paper. Your readers are unlikely to read this paper unless you make this shorter and concentrate on your message. Methodology was repeated in the results and justification of tests was in the methods etc. I strongly recommend authors reduce the length of the paper; maybe move some of the methodology to a supplementary file, simplify the introduction, results and conclusion as the message is lost in the long paper. The discussion is very confusing as it flits from one topic to the next with no cohesion.
Response: The manuscript was thoroughly revised to improve readability. As the reviewer did not specify which parts were repetitive or hard to follow, we did our best to reason how to make the manuscript less wordy.
Most of the Methods were moved to Supplementary Methods; Results were shortened, and repetitive information was deleted; several passages of the discussion were removed and reworded (lines 545-551; 553-560; 587-595; 607-613; 645-651) A conclusion section was included, addressing the main message of the paper (lines 735-750).
I think there are major limitations to this paper, the first is the sample size,(…)
Response: We agree with the reviewer that the main limitation of our study is the small sample size. In fact, the original manuscript addressed this limitation. In this particular study, samples from women who developed gestational diabetes mellitus, pre-eclampsia or other pregnancy complications were not included in the analysis, and this impacted the sample size. On the other hand, we are convinced that our study is based on a well-characterized cohort and that the homogeneity of placental samples contributed to reduce possible confounders of obesity-associated comorbidities. Further comments regarding this limitation have been included in the discussion (547-551; 710-712) of the revised manuscript.
(…)the multiple variables measured (you only used multiple regression in a very small sample size - with so many variables; it can be a ‘shotgun regression’. I suggest you consider other analysis for example AIC )
Response: We agree with the reviewer that the statistical models in our manuscript should not be used quantitatively to estimate the weight of dependent variables (because of the limited sample size). Therefore, we appreciate this comment and we have balanced these data and their interpretation in the manuscript so that the models were not addressed quantitatively (please see lines 511-520 and 681-691, and Table 2). The stepwise backward regression analysis that was used helped to clarify the association of each predictor in the models with the dependent variables, independently of the other factors included in the analysis matrix, and this is a clear advantage over correlation analysis.
It would not be reasonable to consider these results as ‘shotgun’ statistics, as this is defined as a “research in which the investigator studies a large number of variables with no clear strategy or theoretical justification” (Dictionary of Statistics & Methodology, W. Paul Vogt (ed.), 2005, 3rd Ed., DOI: 10.4135/9781412983907.n1795), and this is clearly not the case for these data, as there was a clear strategy (observational clinical trial) and plausible biochemical mechanisms to explain the results. Therefore, we decided to balance this point in the manuscript making the interpretation qualitative, considering mostly the direction of the association of each of the independent variables to the dependent one, i.e., positive or negative association.
Concerning the criteria to keep the independent variables (predictors) in the model, they were clearly described in the original manuscript (lines 313-318 of the revised manuscript). Backward stepwise regression is a built-in feature of the software package used (Statgraphics Centurion v. 18), and it is widely used in the area of nutritional science (for instance, in the following references: Namjoshi SS et al. (2018) Nutrition Deficiencies in Children With Intestinal Failure Receiving Chronic Parenteral Nutrition. Journal of Parenteral and Enteral Nutrition, 42(2): 427-435. DOI: https://doi.org/10.1177/0148607117690528; Yoshida N et al. (2017) Preoperative controlling nutritional status (CONUT) is useful to estimate the prognosis after esophagectomy for esophageal cancer. Langenbecks Arch Surg, 402:333–341. DOI: 10.1007/s00423-017-1553-1; Peters R et al. Nutrition transition, overweight and obesity among rural-to-urban migrant women in Kenya (2019). Public Health Nutrition, 22(17): 3200-3210. DOI: https://doi.org/10.1017/S1368980019001204).
It is expected that any statistical criterion has limitations, and we reasoned the potential limitations of the approach we used by considering the plausible biochemical mechanisms underlying the associations found, and by interpreting the results only qualitatively.
(…) the fact that it is observational and no diet diaries are used. A RCT with supplementation/ no supplementation in lean and obese woman is needed to see if the results are due to intake or disordered metabolism.
Response: We thank the reviewer for raising this point. Dietary records were collected and nutritional counselling was provided to pregnant women as a part of their prenatal care. A detailed analysis of their dietary intake will be the focus of another manuscript, and that is the reason we opted to not include dietary intake data in the present study. That said, we agree with the reviewer regarding the importance of dietary records for data interpretation and we included a table with energy and carbohydrate, protein and lipid intake (Supplementary table 2) in the revised manuscript. There were no significant differences in energy intake or macronutrient intake between groups. Therefore, altered placental metabolism is unlikely to be due to major differences in their diet.
Information of this nature has been included in the Supplementary Methods; Supplementary table 2; Results (lines 331-335) and discussion (lines 557-559 and 725-730) of the revised manuscript.
Reviewer 3 Report
Comments on manuscript nutrients-1329500
In the present manuscript, the authors have investigated an interesting topic related to the placental lipid metabolism and handling from women with pre-gestational obesity as a sole pregnancy complication and compared to placental responses of lean women.
I would like to congratulate Authors for the good-quality of their article, the literature reported used to write the paper, and for the clear and appropriate structure.
The manuscript is well written, presented and discussed, and understandable to a specialist readership.
In general, the organization and the structure of the article are satisfactory and in agreement with the journal instructions for authors. The subject is adequate with the overall journal scope. The work presents a robust study in which a very exhaustive discussion of the literature available has been carried out.
The introduction provides sufficient background, and the other sections include results clearly presented and analyzed.
However, as specific comment, with the aim to further improve the quality of the manuscript, the following changes are suggested:
- Try to insert some exact/numerical values from the most significant results in the Abstract section;
- All acronyms used have to be spelled at the first use;
- A Conclusion section should be inserted to summarize and highlight the obtained results;
So, I recommend the acceptance of the paper after minor revision.
Author Response
Please find below the responses to the reviewers’ comments to the manuscript “Decreased fatty acid transporter FABP1 and increased isoprostanes and neuroprostanes in the human term placenta: implications to inflammation and birth weight in maternal pre-gestational obesity”, by Livia Belcastro, Carolina S. Ferreira, Marcelle A. Saraiva, Daniela B. Mucci, Antonio Murgia, Carla Lai, Claire Vigor, Camille Oger, Jean-Marie Galano, Gabriela D.A. Pinto, Julian L. Griffin, Alexandre G. Torres, Thierry Durand, Graham J. Burton, Fatima L.C. Sardinha and myself.
The manuscript was thoroughly revised for English language by a native speaker. We would like to take the opportunity to thank the reviewers for their careful and thoughtful comments which certainly contributed to improve the quality of the manuscript.
Hoping to hear from you soon
Yours Sincerely,
Tatiana El-Bacha
Associate Professor
Institute of Nutrition
Federal University of Rio de Janeiro
Reviewer 3
Comments on manuscript nutrients-1329500
In the present manuscript, the authors have investigated an interesting topic related to the placental lipid metabolism and handling from women with pre-gestational obesity as a sole pregnancy complication and compared to placental responses of lean women.
I would like to congratulate Authors for the good-quality of their article, the literature reported used to write the paper, and for the clear and appropriate structure.
The manuscript is well written, presented and discussed, and understandable to a specialist readership.
In general, the organization and the structure of the article are satisfactory and in agreement with the journal instructions for authors. The subject is adequate with the overall journal scope. The work presents a robust study in which a very exhaustive discussion of the literature available has been carried out.
The introduction provides sufficient background, and the other sections include results clearly presented and analyzed.
However, as specific comment, with the aim to further improve the quality of the manuscript, the following changes are suggested:
Try to insert some exact/numerical values from the most significant results in the Abstract section;
All acronyms used have to be spelled at the first use;
A Conclusion section should be inserted to summarize and highlight the obtained results;
So, I recommend the acceptance of the paper after minor revision.
Response: We would like to thank the reviewer for carefully reading the manuscript and we are pleased it was appreciated. All the changes suggested by the reviewer were made in the revised version of the manuscript, as follows:
- Abstract: actual values were included for the main set of results (lines 52-56)
- Acronyms were thoroughly revised
- A Conclusion section was included, highlighting the main message of the study (lines 735-750)
Round 2
Reviewer 2 Report
Thank you for revising your paper- it was much easier to read and I was able to more clearly see what the point of the study was and what the results were. It was also useful to define the inclusion criteria and the inclusion of dietary records was a good idea. The limitations of the study are better addressed. It needs be clear that this is a hypothesis generating study.